# BOOSTED RESIDUAL NETWORKS

**Alan Mosca & George D. Magoulas**
Department of Computer Science and Information Systems
Birkbeck, University of London
Malet Street, WC1E 7HX, London, UK
`{a.mosca,gmagoulas}@dcs.bbk.ac.uk`

## ABSTRACT

In this paper we present a new ensemble method, called Boosted Residual Networks, which builds an ensemble of Residual Networks by growing the member network at each round of boosting. The proposed approach combines recent developements in Residual Networks - a method for creating very deep networks by including a shortcut layer between different groups of layers - with the Deep Incremental Boosting, which has been proposed as a methodology to train fast ensembles of networks of increasing depth through the use of boosting. We demonstrate that the synergy of Residual Networks and Deep Incremental Boosting has better potential than simply boosting a Residual Network of fixed structure or using the equivalent Deep Incremental Boosting without the shortcut layers.

## 1 INTRODUCTION

Residual Networks, a type of deep network recently introduced in He et al. (2015a), are characterized by the use of *shortcut* connections (sometimes also called *skip* connections), which connect the input of a layer of a deep network to the output of another layer positioned a number of levels "above" it. The result is that each one of these shortcuts shows that networks can be build in *blocks*, which rely on both the output of the previous layer and the previous block.

Residual Networks have been developed with many more layers than traditional Deep Networks, in some cases with over 1000 blocks, such as the networks in He et al. (2016). A recent study in Veit et al. (2016) compares Residual Networks to an ensemble of smaller networks. This is done by unfolding the shortcut connections into the equivalent tree structure, which closely resembles an ensemble. An example of this can be shown in Figure 1.

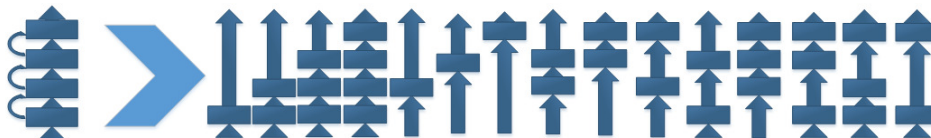

Figure 1: A Residual Network of $N$ blocks can be unfolded into an ensemble of $2^N - 1$ smaller networks.

Dense Convolutional Neural Networks Huang et al. (2016) are another type of network that makes use of shortcuts, with the difference that each layer is connected to all its ancestor layers directly by a shortcut. Similarly, these could be also unfolded into an equivalent ensemble.

True ensemble methods are often left as an *afterthought* in Deep Learning models: it is generally considered sufficient to treat the Deep Learning method as a "black-box" and use a well-known generic Ensemble method to obtain marginal improvements on the original results. Whilst this is an effective way of improving on existing results without much additional effort, we find that it can amount to a waste of computations. Instead, it would be much better to apply an Ensemble method that is aware, and makes us of, the underlying Deep Learning algorithm's architecture.

We define such methods as "white-box" Ensembles, which allow us to improve on the generalisation and training speed compared to traditional Ensembles, by making use of particular properties of the

base classifier's learning algorithm and architecture. We propose a new such method, which we call Boosted Residual Networks, which makes use of developments in Deep Learning, previous other white-box Ensembles and combines several ideas to achieve improved results on benchmark datasets.

Using a white-box ensemble allows us to improve on the generalisation and training speed by making use of the knowledge of the base classifier's structure and architecture. Experimental results show that Boosted Residual Networks achieves improved results on benchmark datasets.

The next section presents the background on Deep Incremental Boosting. Then the proposed Boosted Residual Networks method is described. Experiments and results are discussed next, and the paper ends with conlusions.

## 2 BACKGROUND

Deep Incremental Boosting, introduced in Mosca & Magoulas (2016a), is an example of such white-box ensemble method developed for building ensembles Convolutional Networks. The method makes use of principles from transfer of learning, like for example those used in Yosinski et al. (2014), applying them to conventional AdaBoost (Schapire (1990)). Deep Incremental Boosting increases the size of the network at each round by adding new layers at the end of the network, allowing subsequent rounds of boosting to run much faster. In the original paper on Deep Incremental Boosting Mosca & Magoulas (2016a), this has been shown to be an effective way to learn the *corrections* introduced by the emphatisation of learning mistakes of the boosting process. The argument as to why this works effectively is based on the fact that the datasets at rounds $s$ and $t + 1$ will be *mostly similar*, and therefore a classifier $h_t$ that performs better than randomly on the resampled dataset $X_t$ will also perform better than randomly on the resampled dataset $X_{t+1}$. This is under the assumption that both datasets are sampled from a common ancestor set $X_a$. It is subsequently shown that such a classifier can be re-trained on the differences between $X_t$ and $X_{t+1}$.

This practically enables the ensemble algorithm to train the subsequent rounds for a considerably smaller number of epochs, consequently reducing the overall training time by a large factor. The original paper also provides a conjecture-based justification for why it makes sense to extend the previously trained network to learn the "corrections" taught by the boosting algorithm. A high level description of the method is shown in Algorithm 1, and the structure of the network at each round is illustrated in Figure 3.

---

**Algorithm 1** Deep Incremental Boosting

$\quad D_0(i) = 1/M$ for all $i$
$\quad t = 0$
$\quad W_0 \leftarrow$ randomly initialised weights for first classifier
$\quad$**while** $t < T$ **do**
$\quad\quad X_t \leftarrow$ pick from original training set with distribution $D_t$
$\quad\quad u_t \leftarrow$ create untrained classifier with additional layer of shape $L_{new}$
$\quad\quad$ copy weights from $W_t$ into the bottom layers of $u_t$
$\quad\quad h_t \leftarrow$ train $u_t$ classifier on current subset
$\quad\quad W_{t+1} \leftarrow$ all weights from $h_t$
$\quad\quad \epsilon_t = \frac{1}{2} \sum_{(i,y) \in B} D_t(i)(1 - h_t(x_i, y_i) + h_t(x_i, y))$
$\quad\quad \beta_t = \epsilon_t/(1 - \epsilon_t)$
$\quad\quad D_{t+1}(i) = \frac{D_t(i)}{Z_t} \cdot \beta^{(1/2)(1+h_t(x_i,y_i)-h_t(x_i,y))}$
$\quad\quad$ where $Z_t$ is a normalisation factor such that $D_{t+1}$ is a distribution
$\quad\quad \alpha_t = \frac{1}{\beta_t}$
$\quad\quad t = t + 1$
$\quad$**end while**
$\quad H(x) = \text{argmax}_{y \in Y} \sum_{t=1}^{T} log\alpha_t h_t(x, y)$

---

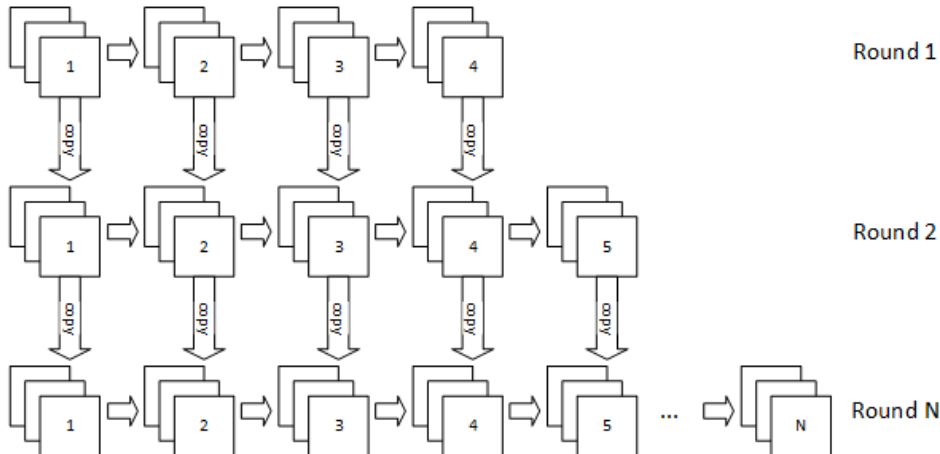

Figure 2: Illusration of subsequent rounds of DIB

# 3 CREATING THE BOOSTED RESIDUAL NETWORK

In this section we propose a method for generating Boosted Residual Networks. This works by increasing the size of an original residual network by one residual block at each round of boosting. The method achieves this by selecting an *injection point* index $p_i$ at which the new block is to be added, which is not necessarily the last block in the network, and by transferring the weights from the layers below $p_i$ in the network trained at the previous round of boosting.

Because the boosting method performs iterative re-weighting of the training set to skew the resample at each round to *emphasize* the training examples that are harder to train, it becomes necessary to utilise the entire ensemble at test time, rather than just use the network trained in the last round. This has the effect that the Boosted Residual Networks cannot be used as a way to train a single Residual Network incrementally. However, as we will discuss later, it is possible to alleviate this situation by deriving an approach that uses bagging instead of boosting; therefore removing the necessity to use the entire ensemble at test time. It is also possible to delete individual blocks from a Residual Network at training and/or testing time, as presented in He et al. (2015a), however this issue is considered out of the scope of this paper.

The iterative algorithm used in the paper is shown in Algorithm 2. At the first round, the entire training set is used to train a network of the original *base* architecture, for a number of epochs $n_0$. After the first round, the following steps are taken at each subsequent round $t$:

- The ensemble constructed so far is evaluated on the training set to obtain the set errors $\epsilon$, so that a new training set can be sampled from the original training set. This is a step common to all boosting algorithms.
- A new network is created, with the addition of a new block of layers $B_{new}$ immediately after position $p_t$, which is determined as an initial pre-determined position $p_0$ plus an offset $i * \delta_p$ for all the blocks added at previous layers. This puts the new block of layers immediately after the block of layers added at the previous round, so that all new blocks are effectively added sequentially.
- The weights from the layers below $p_t$ are copied from the network trained at round $t - 1$ to the new network. This step allows to considerably shorten the training thanks to the transfer of learning shown in Yosinski et al. (2014).
- The newly created network is subsequently trained for a reduced number of epochs $n_{t>0}$.
- The new network is added to the ensemble following the traditional rules and weight $\alpha_t$ used in AdaBoost.

Figure 3 shows a diagram of how the Ensemble is constructed by deriving the next network at each round of boosting from the network used in the previous round.

---

**Algorithm 2** Boosted Residual Networks

$D_0(i) = 1/M$ for all $i$
$t = 0$
$W_0 \leftarrow$ randomly initialised weights for first classifier
set initial injection position $p_0$
**while** $t < T$ **do**
 $X_t \leftarrow$ pick from original training set with distribution $D_t$
 $u_t \leftarrow$ create untrained classifier with an additional block $B_{new}$ of pre-determined shape $N_{new}$
 determine block injection position $p_t = p_{t-1} + |B_{new}|$
 connect the input of $B_{new}$ to the output of layer $p_t - 1$
 connect the output of $B_{new}$ and of layer $p_t - 1$ to a merge layer $m_i$
 connect the merge layer to the remainder of the network
 copy weights from $W_t$ into the bottom layers $l < p_t$ of $u_t$
 $h_t \leftarrow$ train $u_t$ classifier on current subset
 $W_{t+1} \leftarrow$ all weights from $h_t$
 $\epsilon_t = \frac{1}{2} \sum_{(i,y) \in B} D_t(i)(1 - h_t(x_i, y_i) + h_t(x_i, y))$
 $\beta_t = \epsilon_t/(1 - \epsilon_t)$
 $D_{t+1}(i) = \frac{D_t(i)}{Z_t} \cdot \beta^{(1/2)(1 + h_t(x_i, y_i) - h_t(x_i, y))}$
 where $Z_t$ is a normalisation factor such that $D_{t+1}$ is a distribution
 $\alpha_t = \frac{1}{\beta_t}$
 $t = t + 1$
**end while**
$H(x) = \text{argmax}_{y \in Y} \sum_{t=1}^{T} log\alpha_t h_t(x, y)$

---

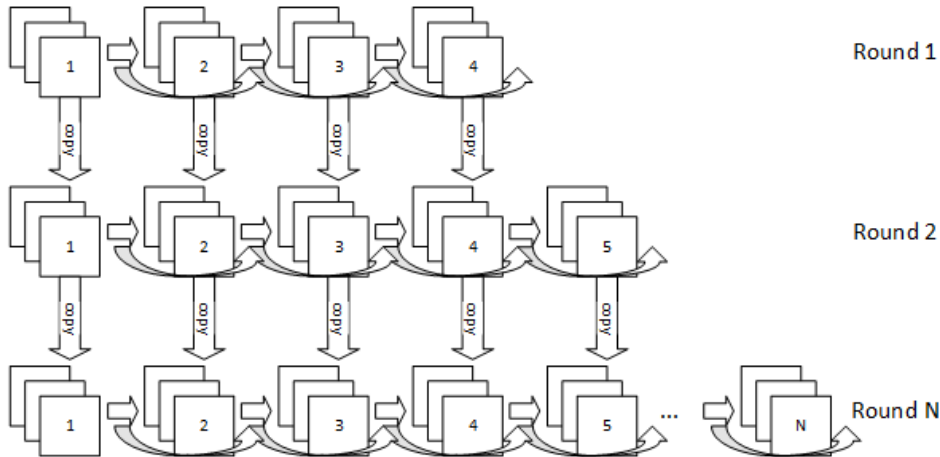

Figure 3: Illusration of subsequent rounds of BRN

We identified a number of optional variations to the algorithm that may be implemented in practice, which we have empirically established as not having an impact on the overall performance of the network. We report them here for completeness.

- Freezing the layers that have been copied from the previous round.

- Only utilising the weights distribution for the examples in the training set instead of resampling, as an input to the training algorithm.

- Inserting the new block always at the same position, rather than after the previously-inserted block (we found this to affect performance negatively).

## 3.1 COMPARISON TO APPROXIMATE ENSEMBLES

While both Residual Networks and Densely Connected Convolutional Networks may be unfolded into an equivalent ensemble, we note that there is a differentiation between an actual ensemble method and an ensemble "approximation". During the creation of an ensemble, one of the principal factors is the creation of *diversity*: each base learner is trained independently, on variations (resamples in the case of boosting algorithms) of the training set, so that each classifier is guaranteed to learn a different function that represents an approximation of the training data. This is the enabling factor for the ensemble to perform better in aggregate.

In the case of Densely Connected Convolutional Networks (DCCN) specifically, one may argue that a partial unfolding of the network could be, from a schematic point of view, very similar to an ensemble of incrementally constructed Residual Networks. We make the observation that, although this would be correct, on top of the benefit of diversity, our method also provides a much faster training methodology: the only network that is trained for a full schedule is the network created at the first round, which is also the smallest one. All subsequent networks are trained for a much shorter schedule, saving a considerable amount of time. Additionally, while the schematic may seem identical, there is a subtle difference: each member network outputs a classification of its own, which is then aggregated by weighted averaging, whilst in a DCCN the input of the final aggregation layer is the output of each underlying set of layers. We conjecture that this aggressive dimensionality reduction before the aggregation will have a regularising effect on the ensemble.

## 4 EXPERIMENTS AND DISCUSSION

|  | Single Net | AdaBoost | DIB | BRN |
|---|---|---|---|---|
| MNIST | 99.41 % | 99.41 % | 99.47 % | 99.53 % |
| CIFAR-10 | 89.12 % | 89.74 % | 90.83 % | 90.85 % |
| CIFAR-100 | 67.25 % | 68.18 % | 68.56 % | 69.04 % |

Table 1: Test accuracy in the three bencharks for the methods compared.

In the experiments we used the MNIST, CIFAR-10 and CIFAR-100 datasets, and compared Boosted Residual Networks (BRN) with an equivalent Deep Incremental Boosting (DIB) without the skip-connections, AdaBoost with the equivalent Residual Network as its base classifier (AdaBoost), and the single Residual Network (Single Net) In order to reduce noise, we aligned the random initialisation of all networks across experiments, by fixing the seeds for the random number generators, and no dataset augmentation was used, both online and offline. Results are reported in Table 1, while Figure 4 shows a side-by-side comparison of accuracy levels at each round of boosting for both DIB and BRN on the MNIST and CIFAR-100 test sets. This figure illustrates how BRNs are able to consistently outperform DIB, regardless of ensemble size, and although such differences still fall within a Bernoulli confidence interval of $95\%$, we make the note that this does not take account of the fact that all the random initialisations were aligned, so both methods started with the exact same network.

Table 2 shows that this is achieved without significant changes in the training time[1]. The main speed increase is due to the fact that the only network being trained with a full schedule is the first network, which is also the smallest, whilst all other derived networks are trained for a much shorter schedule (in this case only $10\%$ of the original training schedule).

The initial network architectures for the first round of boosting are shown in Table 3a for MNIST, and Table 3b for CIFAR-10 and CIFAR-100. It is worth mentioning that we used relatively simple network architectures that were fast to train, which still perform well on the datasets at hand, with accuracy close to, but not comparable to, the state-of-the-art. This enabled us to test larger Ensembles within an acceptable training time.

Training used the WAME method (Mosca & Magoulas (2016b)), which has been shown to be faster than Adam and RMSprop, whilst still achieving comparable generalisation. This is thanks to a

---

[1]In some cases BRN is actually faster than DIB, but we believe this to be just noise due to external factors such as system load.

|          | ResNet  | AdaBoost | DIB     | BRN     |
|----------|---------|----------|---------|---------|
| MNIST    | 115 min | 442 min  | 202 min | 199 min |
| CIFAR-10 | 289 min | 1212 min | 461 min | 449 min |
| CIFAR-100| 303 min | 1473 min | 407 min | 448 min |

Table 2: Training times comparison

| 64 conv, $5 \times 5$ |
|---|
| $2 \times 2$ max-pooling |
| 128 conv, $5 \times 5$ |
| $2 \times 2$ max-pooling * |
| Dense, 1024 nodes |
| 50% dropout |

(a) MNIST

| $2 \times 96$ conv, $3 \times 3$ |
|---|
| 96 conv, $3 \times 3$, $2 \times 2$ strides |
| 96 conv, $3 \times 3$, $2 \times 2$ strides |
| 96 conv, $3 \times 3$, $2 \times 2$ strides |
| $2 \times 2$ max-pooling |
| $2 \times 192$ conv, $3 \times 3$ |
| 192 conv, $3 \times 3$, $2 \times 2$ strides |
| 192 conv, $3 \times 3$, $2 \times 2$ strides |
| 192 conv, $3 \times 3$, $2 \times 2$ strides |
| $2 \times 2$ max-pooling * |
| 192 conv, $3 \times 3$ |
| 192 conv, $1 \times 1$ |
| 10 conv, $1 \times 1$ |
| global average pooling |
| 10-way softmax |

(b) CIFAR-10 and CIFAR-100

Table 3: Network structures used in experiments. The layers marked with "*" indicate the location after which we added the residual blocks.

specific weight-wise learning rate acceleration factor that is determined based only on the sign of the current and previous partial derivative $\frac{\partial E(x)}{\partial w_{ij}}$. For the single Residual Network, and for the networks in AdaBoost, we trained each member for 100 epochs. For Deep Incremental Boosting and Boosted Residual Networks, we trained the first round for 50 epochs, and every subsequent round for 10 epochs, and ran all the algorithms for 10 rounds of boosting, except for the single network. The structure of each incremental block added to Deep Incremental Boosting and Boosted Residual Networks at each round is shown in Table 4a for MNIST, and in Table 4b for CIFAR-10 and CIFAR-100. All layers were initialised following the reccommendations in He et al. (2015b).

**Distilled Boosted Residual Network: DBRN** In another set of experiments we tested the performance of a Distilled Boosted Residual Network (DBRN). Distillation has been shown to be an effective process for regularising large Ensembles of Convolutional Networks in Mosca & Magoulas (2016c), and we have applied the same methodology to the proposed Boosted Residual Network. For the distilled network structure we used the same architecture as that of the Residual Network from the final round of boosting. Accuracy results in testing are presented in Table 5, and for completeness of comparison we also report the results for the distillation of DIB, following the same procedure, as DDIB.

| 64 conv, $3 \times 3$ |
|---|
| Batch Normalization |
| ReLu activation |

(a) MNIST

| 192 conv, $3 \times 3$ |
|---|
| Batch Normalization |
| ReLu activation |
| 192 conv, $3 \times 3$ |
| Batch Normalization |
| ReLu activation |

(b) CIFAR-10 and CIFAR-100

Table 4: Structure of blocks added at each round of DIB and BRN.

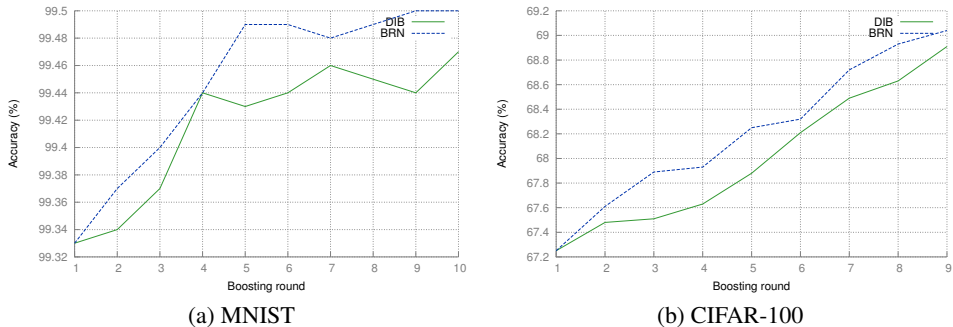

(a) MNIST (b) CIFAR-100

Figure 4: Round-by-round comparison of DIB vs BRN on the test set

|  | DBRN | DDIB |
|---|---|---|
| MNIST | 99.49 % | 99.44 % |
| CIFAR-10 | 91.11 % | 90.66 % |
| CIFAR-100 | 66.63 % | 65.91 % |

Table 5: Comparative results in terms of testing accuracy.

**Bagged Residual Networks: BARN**  We experimented with substituting the boosting algorithm with a simpler bagging algorithm (Breiman (1996)) to evaluate whether it would be possible to only use the network from the final round of bagging as an approximation of the Ensemble. We called this the Bagged Approximate Residual Networks (BARN) method. We then also tested the performance of the Distilled version of the whole Bagging Ensemble for comparison. The results are reported as "DBARN". The results are reported in Table 6. It is clear that trying to use the last round of bagging is not comparable to using the entire Bagging ensemble at test time, or deriving a new distilled network from it.

## 5  CONCLUSIONS AND FUTURE WORK

In this paper we have derived a new ensemble algorithm specifically tailored to Convolutional Networks to generate Boosted Residual Networks. We have shown that this surpasses the performance of a single Residual Network equivalent to the one trained at the last round of boosting, of an ensemble of such networks trained with AdaBoost, and Deep Incremental Boosting on the MNIST and CIFAR datasets, without using augmentation techniques.

We then derived and looked at a distilled version of the method, and how this can serve as an effective way to reduce the test-time cost of running the Ensemble. We used Bagging as a proxy to test generating the approximate Residual Network, which, with the parameters tested, does not perform as well as the original Residual Network, BRN or DBRN.

Further experimentation of the Distilled methods presented in the paper, namely DBRN and DBARN, is necessary to fully investigate their behaviour. This is indeed part of our work in the near future. Additionally, the Residual Networks built in our experiments were comparatively smaller than those that achieve state-of-the-art performance. Reaching state-of-the-art on specific benchmark datasets was not our goal, instead we intended to show that we developed a methodology that makes it feasible to created ensembles of Residual Networks following a "white-box" approach to

|  | BRN | Bagging | BARN | DBARN |
|---|---|---|---|---|
| MNIST | 99.50 % | 99.55 % | 99.29 % | 99.36 % |
| CIFAR-10 | 90.56 % | 91.43 % | 88.47 % | 90.63 % |
| CIFAR-100 | 69.04 % | 68.15 % | 69.42 % | 66.16 % |

Table 6: Test accuracy for BARN.

significantly improve the training times and accuracy levels. Nevertheless, it might be appealing in the future to evaluate the performance improvements obtained when creating ensembles of larger, state-of-the-art, networks. Additional further investigation could also be conducted on the creation of Boosted Densely Connected Convolutional Networks, by applying the same principle to DCCN instead of Residual Networks.

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
