# Peer review of "Boosted Residual Networks"

_ICLR 2017 — rejected_

[Reviewer Comment · AnonReviewer2 · 04 Dec 2016]
**comparisons and setup**

- Can you give the details of the experiment setup e.g. parameters to be tuned, algorithm to train at each step of boosting etc? Also can you give the details of networks architecture and references? 
- Can you elaborate on comparison to state of resNet variants, dense convolutional network?
- Can you give also comparison on training time?
- Do you have any result on Imagenet?

[Official Review · AnonReviewer1 · rating 3 · confidence 5 · 16 Dec 2016]
**Interesting ideas, unconvincing execution, lack of comparisons to the literature**

The paper under consideration proposes a set of procedures for incrementally expanding a residual network by adding layers via a boosting criterion.

The main barrier to publication is the weak empirical validation. The tasks considered are quite small scale in 2016 (and MNIST with a convolutional net is basically an uninteresting test by this point). The paper doesn't compare to the literature, and CIFAR-10 results fail to improve upon rather simple, single-network published baselines (Springenberg et al, 2015 for example, obtains 92% without data augmentation) and I'm pretty sure there's a simple ResNet result somewhere that outshines these too. The CIFAR100 results are a little bit interesting as they are better than I'm used to seeing (I haven't done a recent literature crawl), and this is unsurprising -- you'd expect ensembles to do well when there's a dearth of labeled training data, and here there are only a few hundred per label. But then it's typical on both CIFAR10 and CIFAR100 to use simple data augmentation schemes which aren't employed here, and these and other forms of regularization are a simpler alternative to a complicated iterative augmentation scheme like this.

It'd be easier to sell this method either as an option for scarce labeled datasets where data augmentation is non-trivial (but then for most image-related applications, random crops and reflections are easy and valid), but that would necessitate different benchmarks, and comparison against simpler methods like said data augmentation, dropout (especially, due to the ensemble interpretation), and so on.

[Official Review · AnonReviewer2 · rating 3 · confidence 5 · 19 Dec 2016]
**The contribution is incremental with no impressive comparison results**

This paper proposes a boosting based ensemble procedure for residual networks by adopting the Deep Incremental Boosting method that was used for CNN's(Mosca & Magoulas, 2016a). At each step t, a new block of layers are added to the network at a position p_t and the weights of all layers are copied to the current network to speed up training.

The method is not sufficiently novel since the steps of Deep Incremental Boosting are slightly adopted. Instead of adding a layer to the end of the network, this version adds a block of layers to a position p_t (starts at a selected position p_0) and merges layer accordingly hence slightly adopts DIB. 

The empirical analysis does not use any data-augmentation. It is not clear whether the improvements (if there is) of the ensemble disappear after data-augmentation.  Also, one of the main baselines, DIB has no-skip connections therefore this can negatively affect the fair comparison. The authors argue that they did not involve state of art Res Nets since their analysis focuses on the ensemble approach, however any potential improvement of the ensemble can be compensated with an inherent feature of Res Net variant. The boosting procedure can be computationally restrictive in case of ImageNet training and Res Net variants may perform much better in that case too. Therefore the baselines should include the state of art Res Nets and Dense Convolutional networks hence current results are preliminary.

In addition, it is not clear how sensitive the boosting to the selection of injection point.

This paper adopts DIB to Res Nets and provides some empirical analysis however the contribution is not sufficiently novel and the empirical results are not satisfactory for demonstrating that the method is significant.

Pros
-provides some preliminary results for boosting of Res Nets
Cons
-not sufficiently novel: an incremental approach 
-empirical analysis is not satisfactory

[Official Review · AnonReviewer3 · rating 4 · confidence 5 · 19 Dec 2016]
**Lack of comparison**

The authors mention that they are not aiming to have SOTA results.
However, that an ensemble of resnets has lower performance than some of single network results, indicates that further experimentation preferably on larger datasets is necessary.
The literature review could at least mention some existing works such as wide resnets

[Final Decision · Program Chairs · 06 Feb 2017]
**ICLR committee final decision**

All three reviewers point to significant deficiencies. No response or engagement from the authors (for the reviews). I see no basis for supporting this paper.